# Analysis of cathepsin S expression in gastric adenocarcinoma and in *Helicobacter pylori* infection

Adriano C. Costa[1,2⊙]*, Fernando Santa-Cruz[2⊙], Raphael L. C. Araújo[3], Glauber Leitão[1], José-Luiz Figueiredo[4‡], Álvaro A. B. Ferraz[4‡]

**1** Oncology Unit, Hospital das Clínicas, Federal University of Pernambuco (HC-UFPE), Recife, Pernambuco, Brazil, **2** Post-graduation in Surgery, Federal University of Pernambuco, Recife, Pernambuco, Brazil, **3** Department of Digestive Surgery, Universidade Federal de São Paulo (UNIFESP), São Paulo, São Paulo, Brazil, **4** Department of Surgery, Federal University of Pernambuco, Recife, Pernambuco, Brazil

⊙ These authors contributed equally to this work.
‡ JLF and AABF also contributed equally to this work.
* adrianocacosta@gmail.com

**Data Availability Statement:** All relevant data are within the paper.

**Funding:** The author(s) received no specific funding for this work.

## Abstract

### Background

Recent experimental studies have suggested a potential link between cathepsin S (CTTS) and gastric adenocarcinoma progression. Herein, we aimed to evaluate the expression of CTTS in gastric adenocarcinoma in patients who underwent curative-intent surgical resection.

### Methods

This was a cross-sectional study that included two groups: gastric adenocarcinoma (n = 42) and gastritis (n = 50). The gastritis group was then subdivided into H. pylori-positive (n = 25) and H. pylori-negative (n = 25) groups. Gastric tissue samples were analysed to determine CTTS expression through immunohistochemistry. Samples were obtained by oesophago-gastroduodenoscopy or surgical specimens.

### Results

In patients with gastritis, the age ranged from 18 to 78 years. Among them, 34% were male, and 66% were female. In patients with gastric adenocarcinoma, the age ranged from 37 to 85 years. Among them, 50% were male. When comparing the expression of CTTS between the two groups, only 16% of the gastritis samples had an expression higher than 25%. Alternatively, among patients with gastric adenocarcinoma, 19% had expression between 25–50%, 14.3% between 51–75%, and 26.2% had expression higher than 75% (p < 0.001). In the gastritis group, CTTS expression was significantly higher in patients with a positive test for *H. pylori* than negative test for *H. pylori*: 87.5% and 38.5%, respectively (p<0.001). There was no statistically significant association between CTTS positivity and clinicopathological variables, including tumour staging, histological type, angiolymphatic invasion, recurrence, current status and death.

**Competing interests:** The authors have declared that no competing interests exist.

## Conclusion

CTTS expression is higher in gastric adenocarcinoma samples. Patients with gastritis due to *H. pylori* also show a higher expression of CTTS than patients with negative results for this bacterium.

## Introduction

Cathepsins are enzymes (proteases) widely distributed in both intra- and extracellular spaces of diverse tissues of the digestive system, mainly located within lysosomes and other acidic environments [1, 2]. Among its family of 15 lysosomal proteases, at least five (cathepsin B, H, K, L, and S) have been repeatedly associated with cancer progression, specifically for solid tumours [3, 4]. Their mechanisms vary and involve degradation of the extracellular matrix and modification of the tumour microenvironment [5, 6]

In digestive cancers, the expression of cathepsin is positively regulated by tumour-promoting factors, such as C-myc, K-ras, AGR2, MAPK, p38, and the Hedgehog (Hh) signalling pathways [7–9]. Moreover, cathepsins activate growth factors, such as epidermal growth factor (EGF), vascular endothelial growth factor (VEGF) and tumor growth factor-beta (TGFβ), promoting cancer cell proliferation and angiogenesis, and have regulatory properties in apoptosis, thus affecting multiple stages of tumorigenesis [8–11].

Recently, some studies have pointed to a putative relationship between gastric adenocarcinoma and cathepsin expression, suggesting potential therapeutic, prognostic, and diagnostic roles of this enzyme in the evolution of this disease [12]. Moreover, in vitro studies have shown that increased cathepsin S (CTTS) expression is related to increased tumour invasion and metastasis and that its inhibition may prevent tumour cell invasion and migration in gastric adenocarcinoma [13, 14].

This study evaluated the expression of CTTS in gastric tissue samples of patients with gastric adenocarcinoma and compared it with the expression in gastric tissue samples of gastritis patients without cancer.

## Materials and methods

### Study design

A cross-sectional study was performed at Hospital das Clínicas, Federal University of Pernambuco, Recife, Brazil, comparing the expression of CTTS in gastric tissue samples of patients diagnosed with gastric adenocarcinoma (n = 42) and patients diagnosed only with gastritis (n = 50). The samples were obtained by oesophagogastroduodenoscopy (EGD) or by surgical specimen. After, the group of patients with gastritis was subdivided into two subgroups: one with a positive result for *H. pylori* (n = 25) and other with a negative result for *H. pylori* (n = 25). The primary endpoint was to compare CTTS expression assessed by immunohistochemistry in gastric tissue samples from patients with adenocarcinoma and patients with gastritis (with and without *H. pylori*). This study was performed in accordance with the institutional review board and its policy for protected health information.

All procedures performed in this study involving human participants were in accordance with the ethical standards of the institutional research committee and the 1964 Helsinki declaration and its later amendments or comparable ethical standards. This research protocol was approved by the Ethics Committee of the Hospital das Clínicas da Universidade Federal de

Pernambuco (HC/UFPE-EBSERH) under protocol CAAE no. 38000620.9.0000.8807. Informed consent was obtained from all participants in the study as outlined in PLOS consent form.

## Selection of patients

We included patients who underwent surgical curative-intent treatment for gastric adenocarcinoma. Patients at stage IV and those undergoing neoadjuvant chemotherapy were excluded. The control group was formed by consecutive patients presenting a confirmed histopathological result for gastritis in the pathology obtained by EGD. The search for *H. pylori* was performed in all patients using the urease test and confirmed by histopathology with Giemsa staining. Patients previously submitted to gastroplasty and those with reports of previous treatment for *H. pylori* were excluded.

## Immunohistochemistry

Immunohistochemical (IHC) staining was performed to study the expression of CTTS in both groups. The 3-μm sections were used in series for IHC analysis and placed on Superfrost Plus glass slides. The immunostaining used was the Ventana BenchMark ULTRA System automated staining system using rabbit polyclonal antibody directed against CTTS (Clone No. A13482; ABclonal, Massachusetts, United States). A 1:100 dilution was used and incubated for 30 min at 37 ˚C. The DAB IHC detection kit was used as the chromogen substrate. All specimens were counterstained with haematoxylin. The IHC reactions were interpreted using a standard optical microscope and analysed according to the specific pattern of the investigated antibody. The marking intensity was assessed using the following grading: 0 if no detectable colouring; 1 if weak colouring (light yellow); 2 if moderate colouring (brown-yellow); 3 if strong colouring (brown).

We graded the percentage of stained cells in both groups as follows: 0 (no positive tumour cells); 1 (1–25% of positive tumour cells); 2 (26–50% of positive tumour cells); 3 (51–75% of positive tumour cells); 4 (>75% of positive tumour cells).

The staining index score was calculated as the product of the percentage of positive tumour cells and the intensity of staining. We defined CTTS expression according to the colour index: 0 (negative); 1–4 (weakly positive); 5–8 (positive); 9–12 (strongly positive).

For analysis purposes, the CTTS expression intensity was categorically assessed as high or low expression. We defined high expression as a colour index score >4, while low expression was a score ≤ 4. An index = 0 indicating missing expression.

## Statistical analysis

For statistical analysis, we used STATA/SE 12.0 software (StataCorp, College Station, TX). The results are expressed as the mean values and standard deviations or proportions, as appropriate. A p value < 0.05 was considered significant in all tests. Associations were verified using the chi-square test and Fisher's exact test for categorical variables.

## Results

Among ninety-two (92) patients studied from 2017 to 2019, 50 patients had gastritis, 42 patients had gastric adenocarcinoma, and the clinical demographic data are shown in Table 1. In patients with gastritis, the age ranged from 18 to 78 years. Among them, 35 patients (70%) were under 50 years old, 17 (34%) were male, and 33 (66%) were female. In patients with gastric adenocarcinoma, the age ranged from 37 to 85 years. Among them, 34 patients (81%) were

**Table 1. Comparison between gastric adenocarcinoma and gastritis groups (demographic data).**

| Variable | Group | | | | p value* |
|---|---|---|---|---|---|
| | Gastric adenocarcinoma | | Gastritis | | |
| | n | % | n | % | |
| **Age** | | | | | |
| Under 50 | 8 | 19.0 | 35 | 70.0 | <**0.001** |
| Over 50 | 34 | 81.0 | 15 | 30.0 | |
| **Gender** | | | | | |
| Male | 21 | 50.0 | 17 | 34.0 | 0.121 |
| Female | 21 | 50.0 | 33 | 66.0 | |
| ***H. pylori*** | | | | | |
| Positive | 8 | 19.0 | 24 | 48.0 | **0.004** |
| Negative | 34 | 81.0 | 26 | 52.0 | |
| **Percentage of CTTS stained cells** | | | | | |
| No positive cells | 07 | 16.7 | 19 | 38.0 | <**0.001** |
| 1–25% | 10 | 23.8 | 23 | 46.0 | |
| 25–50% | 08 | 19.0 | 04 | 8.0 | |
| 51–75% | 06 | 14.3 | 02 | 4.0 | |
| >75% | 11 | 26.2 | 02 | 4.0 | |
| **CTTS colouring index** | | | | | |
| Negative (0) | 7 | 16.7 | 19 | 38.0 | **0.002** |
| Weakly positive (1–4) | 14 | 33.3 | 24 | 48.0 | |
| Positive (5–8) | 09 | 21.4 | 03 | 6.0 | |
| Strongly positive (9–12) | 12 | 28.6 | 04 | 8.0 | |
| **Intensity of expression** | | | | | |
| Absent—0 | 7 | 16.7 | 19 | 38.0 | **0.001** |
| Low < 4 | 14 | 33.3 | 24 | 48.0 | |
| High ≥ 4 | 21 | 50.0 | 07 | 14.0 | |

(*) Chi Square Test

over 50 years old, 21 (50%) were male, and 21 (50%) were female. When comparing the expression of CTTS between the two groups, in gastritis samples, 38% did not express CTTS, 46% had low expression (1–25%), and only 16% had an expression higher than 25%. Alternatively, among patients with gastric adenocarcinoma, 19% had expression between 25–50%, 14.3% between 51–75%, and 26.2% had expression higher than 75%, with significant results (p < 0.001). Analyses involving the CTTS staining index and the IHC intensity also showed significance, with expression higher in the group of patients with gastric adenocarcinoma, as demonstrated in Table 1.

Regarding the cancer group, the pathology findings and oncological staging are summarized in Table 2. Most cases presented tumours in the antrum (54.8%), underwent total gastrectomy (52.4%), and were stage IIIB (45.3%), and both intestinal and diffuse Lauren subtypes were equally present (42.9%).

In the evaluation of CTTS expression in the group of patients with gastritis, CTTS expression was significantly higher in patients with a positive test for *H. pylori*: 87.5% and 38.5% (p<0.001), as depicted in Table 3. The IHC staining of CTTS in a gastric tissue sample with gastritis is depicted in Fig 1.

In the evaluation of CTTS expression in the group of patients with gastritis, CTTS expression was significantly higher in patients with a positive test for *H. pylori*: 87.5% and 38.5%

**Table 2. Characterization of the group of patients with gastric cancer.**

| Variable | Gastric cancer group | |
|---|---|---|
| | n | % |
| **Topography** | | |
| Antrum | 23 | 54.8 |
| Body | 19 | 45.2 |
| **Type of surgery** | | |
| Subtotal gastrectomy | 20 | 47.6 |
| Total gastrectomy | 22 | 52.4 |
| **Pathological stage** | | |
| IA | 10 | 23.8 |
| IB | 10 | 23.8 |
| IIIB | 19 | 45.3 |
| IIIC | 03 | 7.1 |
| **Primary tumour** | | |
| T1 | 10 | 23.8 |
| T2 | 10 | 23.8 |
| T3 | 19 | 45.3 |
| T4 | 03 | 7.1 |
| **Lymph nodes** | | |
| N0 | 20 | 47.6 |
| N3 | 22 | 52.4 |
| **Histological type** | | |
| Intestinal | 18 | 42.9 |
| Diffuse | 18 | 42.9 |
| Mixed | 06 | 14.2 |
| **Histological grade** | | |
| Well differentiated | 03 | 7.1 |
| Moderate | 08 | 19.0 |
| Poorly differentiated | 31 | 73.9 |
| **Angiolymphatic invasion** | | |
| Positive | 23 | 54.8 |
| Negative | 19 | 45.2 |
| **Recurrence** | | |
| No | 32 | 76.2 |
| Yes | 10 | 23.8 |
| **Current status** | | |
| Alive without disease | 30 | 71.4 |
| Alive with disease | 03 | 7.1 |
| Death without cancer | 02 | 4.8 |
| Death with cancer | 07 | 16.7 |

($p < 0.001$). The IHC staining of CTTS in a gastric tissue sample with gastritis is depicted in Fig 1.

In the evaluation of CTTS expression in the group of patients with gastric adenocarcinoma, there was no significant association between the positivity of the expression and the clinico-pathological variables, as demonstrated in Table 4, and the IHC staining of CTTS in a gastric cancer tissue sample is shown in Fig 2. The statistical power of this sample was 72.7%

**Table 3. Expression of CTTS in the group of patients with gastritis.**

| Variable | CTTS colouring score | | | | p value* |
|----------|------|------|------|------|----------|
| | Positive | | Negative | | |
| | n | % | n | % | |
| **Age** | | | | | |
| Under 50 | 22 | 62.9 | 13 | 37.1 | 0.849 |
| Over 50 | 09 | 60.0 | 06 | 40.0 | |
| **Gender** | | | | | |
| Male | 11 | 64.7 | 06 | 35.3 | 0.777 |
| Female | 20 | 60.6 | 13 | 39.4 | |
| ***H. pylori*** | | | | | |
| Positive | 21 | 87.5 | 03 | 12.5 | <**0.001** |
| Negative | 10 | 38.5 | 16 | 61.5 | |

(*) Chi Square Test

according to the presence of CTTS expression in the gastric cancer groups compared to benign stomach lesions.

## Discussion

Cathepsins that have previously shown increased expression in the presence of gastric cancer are B, E, K, L, S, X, and Z [8]. To date, only a few studies have sought to assess the relationship between CTTS and gastric cancer [13, 14]. This enzyme appears to play an important role in the tumour invasion process through the degradation of the extracellular matrix, modulation of the immune response, and regulation of several cell signalling pathways, including the activation of tyrosine kinase receptors, especially c-Met, matrix metalloproteinases, IL-11, CXCL16, and integrin alpha-6-beta-4 [4, 13, 15]. Specifically for gastric adenocarcinoma, CTTS appears to have an activating effect on the MKN7 and MKN45 cancer cell lines [13].

Liu et al. [14] evaluated the serum dosage of CTTS in patients with gastric cancer by comparing the results with healthy patients and with benign gastric lesions. They observed that the serum CTTS values of patients with gastric cancer were significantly higher than those of non-tumour gastric tissue controls (P < 0.001). In that study, the authors investigated the diagnostic power of CTTS in 496 patients, finding sensitivity and specificity values of 60.7% and 90%, respectively. Additionally, in that study, there was a significant decrease in serum CTTS levels after surgical resection of the tumour, suggesting an intimate relation between this enzyme and the tumour microenvironment. In our study, we found similar results, with CTTS expression significantly higher in the group of patients with gastric adenocarcinoma than in the control group. The results of these studies suggest that CTTS may be a potential biomarker for the diagnosis of gastric cancer.

Yang et al. [13] studied the expression of cathepsins through a proteomic analysis of cultures of normal cells and gastric cancer cells. We observed higher protein expression and positive regulation of cathepsin S in the gastric cancer cell secretome. There were no statistically significant differences in CTTS expression between the intestinal, diffuse, and mixed subtypes.

Researchers have shown a correlation between CTTS and disease characteristics, such as tumour size, lymph node invasion, distant metastases, and overall survival, noting that higher CTTS expression was related to more advanced TNM stages and worse survival rates [14]. In the present study, there was no statistically significant association between CTTS expression

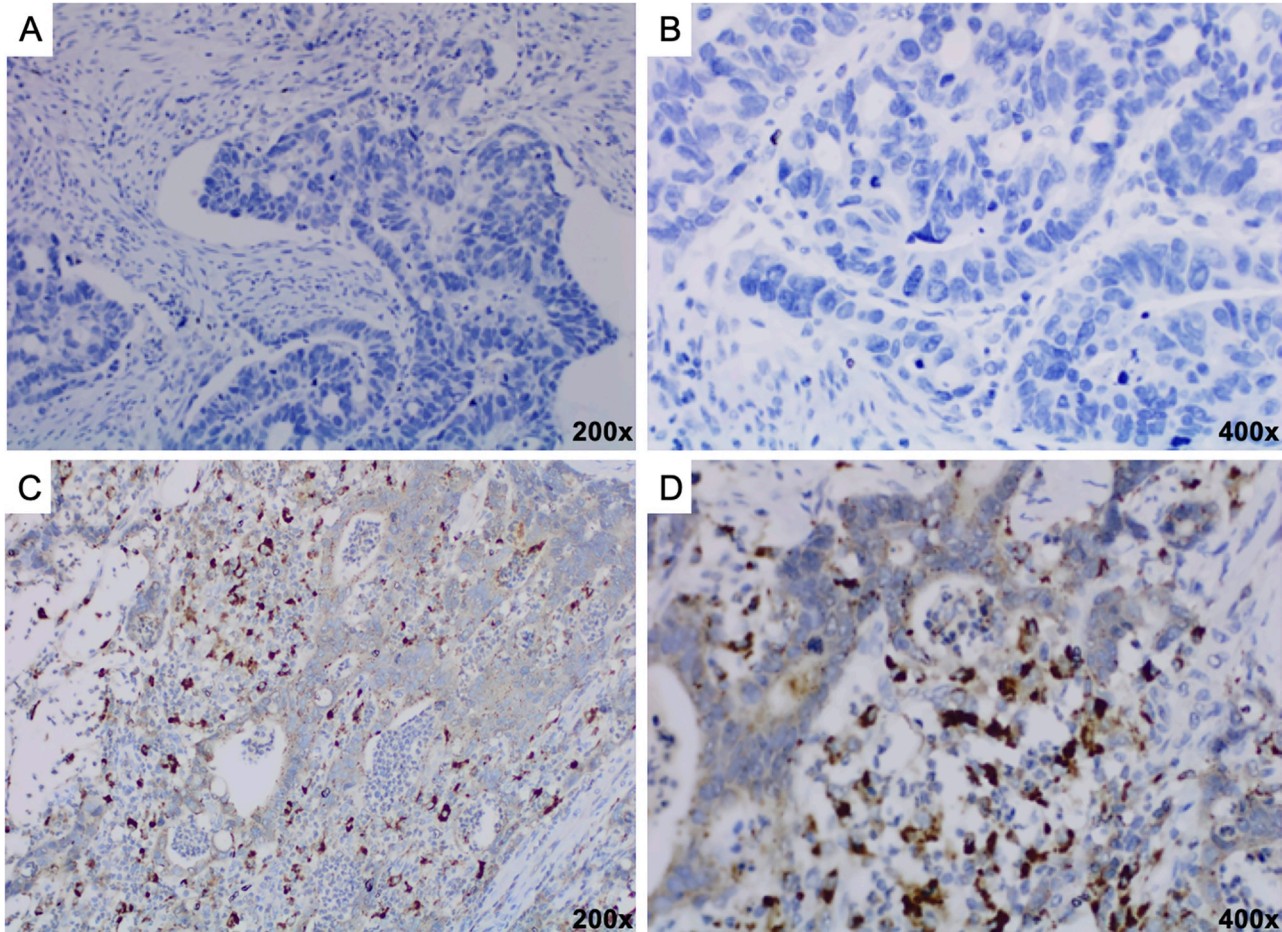

**Fig 1. (A-B): IHC staining showing negative expression of CTTS in a gastric tissue sample with gastritis; (C-D): IHC staining showing positive expression of CTTS in a gastric tissue sample with gastritis.**

and tumour staging or survival rates. A possible explanation for such a difference between the studies is the number of patients included, which was noticeably lower in our analysis.

Infection of the gastric mucosa by *H. pylori* is an important risk factor for the development of gastric adenocarcinoma. However, the exact mechanisms of carcinogenesis activation have not yet been fully elucidated [16]. One of the possible mechanisms noted in this process is the proinflammatory response orchestrated by Th17 cells in the infected gastric mucosa [17, 18]. Previous studies have shown an association between *H. pylori* infection and increased levels of cathepsins D and X. However, there are no studies determining the behaviour of CTTS in the presence of an *H. pylori* infection [19, 20]. In the present study, we evaluated the expression of CTTS in samples of gastric mucosa infected by *H. pylori*. We observed that 87.5% of the samples in the gastritis group with *H. pylori* showed positive expression for CTTS, contrasting with only 12.5% of the gastritis group without *H. pylori*. These results reinforce the hypothesis that CTTS is involved in the process of carcinogenesis of gastric adenocarcinoma, as it also has a higher expression.

This study has some limitations that deserve attention. First, the sample size was limited, due to the single-centre nature of this study. As the sample was nonprobabilistic and selected by convenience, we did not calculate the sample size as we included in the analysis all patients

**Table 4. Expression of CTTS in the group of patients with gastric adenocarcinoma.**

| Variable | CTTS | | | | p value[*] |
|---|---|---|---|---|---|
| | Positive | | Negative | | |
| | n | % | n | % | |
| **Age** | | | | | |
| Under 50 | 7 | 87.5 | 01 | 12.5 | 1.000 |
| Over 50 | 28 | 82.4 | 06 | 17.6 | |
| **Gender** | | | | | |
| Male | 17 | 81.0 | 04 | 19.0 | 1.000 |
| Female | 18 | 85.7 | 03 | 14.3 | |
| **H. pylori** | | | | | |
| Positive | 27 | 79.4 | 7 | 20.6 | 0.312 |
| Negative | 08 | 100.0 | 0 | 0.0 | |
| **Topography** | | | | | |
| Antrum | 18 | 78.3 | 05 | 21.7 | 0.428 |
| Body | 17 | 89.5 | 02 | 10.5 | |
| **Type of surgery** | | | | | |
| Subtotal gastrectomy | 15 | 75.0 | 05 | 25.0 | 0.229 |
| Total gastrectomy | 20 | 90.9 | 02 | 9.1 | |
| **Staging** | | | | | |
| IA | 08 | 80.0 | 02 | 20.0 | 0.490 |
| IB | 07 | 70.0 | 03 | 30.0 | |
| IIIB | 17 | 89.5 | 02 | 10.5 | |
| IIIC | 03 | 100.0 | 0.0 | 0.0 | |
| **Primary tumour** | | | | | |
| T1 | 08 | 80.0 | 02 | 20.0 | 0.490 |
| T2 | 07 | 70.0 | 03 | 30.0 | |
| T3 | 17 | 89.5 | 02 | 10.5 | |
| T4 | 03 | 100.0 | 0.0 | 0.0 | |
| **Lymph nodes** | | | | | |
| N0 | 15 | 75.0 | 05 | 25.0 | 0.229 |
| N3 | 20 | 90.0 | 02 | 9.1 | |
| **Histological type** | | | | | |
| Intestinal | 14 | 77.8 | 04 | 22.2 | 0.852 |
| Diffuse | 16 | 88.9 | 02 | 11.1 | |
| Mixed | 05 | 83.3 | 01 | 16.7 | |
| **Histological grade** | | | | | |
| Well differentiated | 03 | 100.0 | 0 | 0.0 | 0.177 |
| Moderate | 05 | 62.5 | 03 | 37.5 | |
| Poorly differentiated | 27 | 87.1 | 04 | 12.9 | |
| **Angiolymphatic invasion** | | | | | |
| Positive | 20 | 87.0 | 03 | 13.0 | 0.682 |
| Negative | 15 | 78.9 | 04 | 21.1 | |
| **Recurrence** | | | | | |
| No | 26 | 81.3 | 06 | 18.8 | 1.000 |
| Yes | 09 | 90.0 | 01 | 10.0 | |
| **Current status** | | | | | |
| Alive without disease | 24 | 80.0 | 06 | 20.0 | 0.475 |
| Alive with disease | 02 | 66.7 | 01 | 33.3 | |
| Death without cancer | 02 | 100.0 | 0 | 0.0 | |
| Death with cancer | 07 | 100.0 | 0 | 0.0 | |

*(Continued)*

**Table 4.** (Continued)

| Variable | CTTS | | | | p value* |
|---|---|---|---|---|---|
| | Positive | | Negative | | |
| | n | % | n | % | |
| **Death** | | | | | |
| Yes | 09 | 100.0 | 0 | 0.0 | 0.314 |
| No | 26 | 78.8 | 07 | 21.2 | |

(*) Fisher's exact test

operated on during the study period. However, this analysis had enough power to detect a difference between groups. Another limitation of note is related to the observational and cross-sectional nature of this study. A longitudinal study could have provided reliable information about the relationship between CTTS expression and patient survival. However, for our primary endpoint, the methodology applied was adequate.

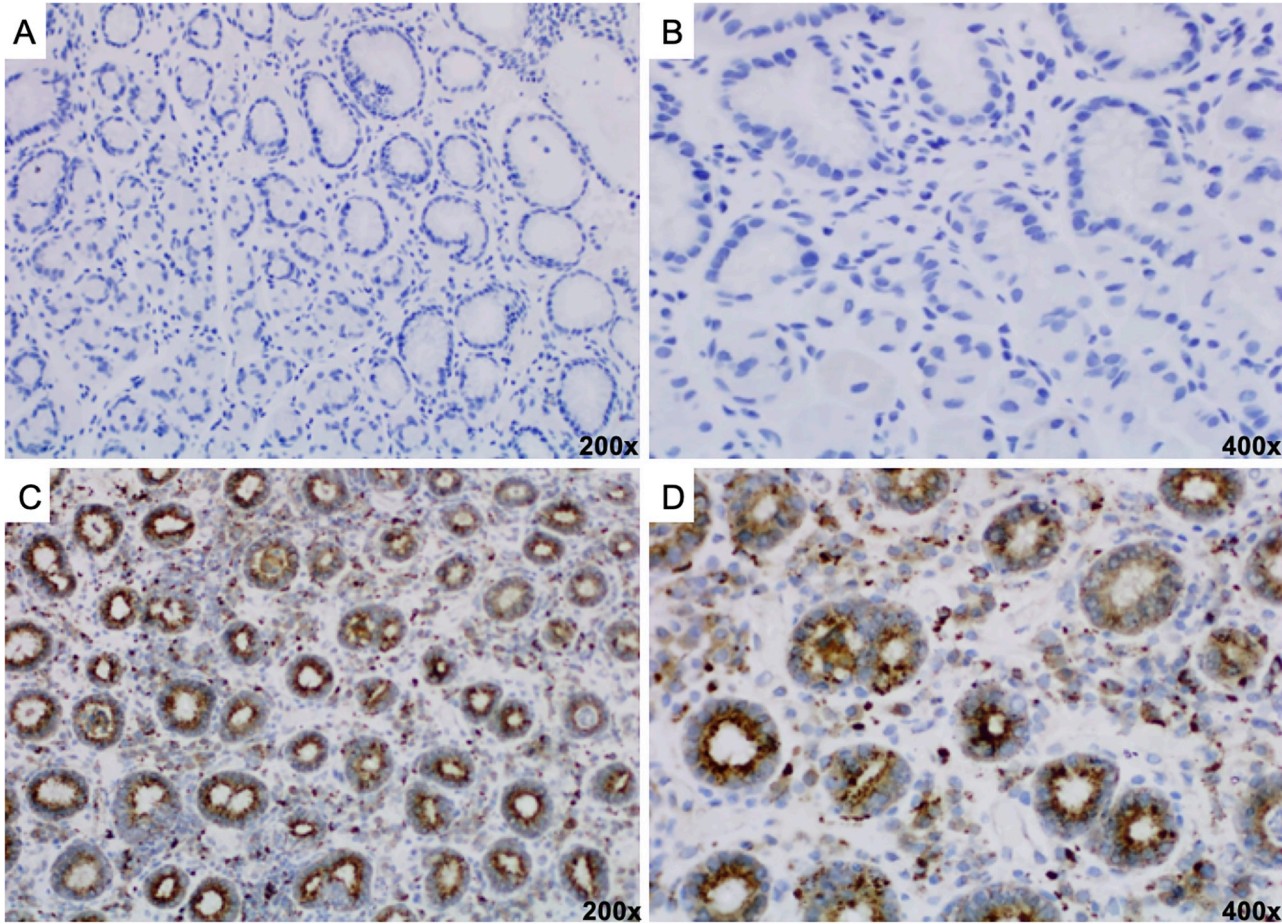

**Fig 2. (A-B): IHC staining showing negative expression of CTTS in gastric adenocarcinoma; (C-D): IHC staining showing positive expression of CTTS in gastric adenocarcinoma.**

In contrast to the limitations discussed above, the present study reports important data that provide robustness and authenticity to the analysis. It is one of the few studies to assess the expression of CTTS in samples of gastric adenocarcinoma in humans and the first to attest to a possible relationship between the expression of this enzyme and infection by *H. pylori*, an important risk factor for the development of gastric adenocarcinoma.

In summary, the results of the present study showed that CTTS has higher expression in gastric adenocarcinoma than in nontumour tissue samples. Moreover, patients with gastritis by *H. pylori* also show a higher expression of CTTS than patients with gastritis with negative results for this bacterium. These results reinforce the discussion about the role of CTTS in the evolution of gastric cancer. Nevertheless, further studies are needed to define the relationship of this enzyme in the process of gastric adenocarcinoma carcinogenesis.

## Acknowledgments

The authors expressed their appreciation to the Anatomopathological Diagnostic Center (CEDAP), João Pessoa, PB, Brazil.

## Author Contributions

**Conceptualization:** José-Luiz Figueiredo, Álvaro A. B. Ferraz.

**Data curation:** Adriano C. Costa, Fernando Santa-Cruz.

**Formal analysis:** Glauber Leitão.

**Investigation:** Adriano C. Costa, Fernando Santa-Cruz.

**Methodology:** Raphael L. C. Araújo, José-Luiz Figueiredo, Álvaro A. B. Ferraz.

**Supervision:** José-Luiz Figueiredo, Álvaro A. B. Ferraz.

**Visualization:** Glauber Leitão.

**Writing – original draft:** Adriano C. Costa, Fernando Santa-Cruz.

**Writing – review & editing:** Raphael L. C. Araújo.

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
