## [Decision Letter · Decision Letter 0]

5 Apr 2022

PONE-D-21-36174Analysis of the expression of cathepsin S in gastric adenocarcinoma and in Helicobacter pylori infectionPLOS ONE

Dear Dr. Adriano Carneiro Costa,

Thank you for submitting your manuscript to PLOS ONE. After careful consideration, we feel that it has merit but does not fully meet PLOS ONE’s publication criteria as it currently stands. Therefore, we invite you to submit a revised version of the manuscript that addresses the points raised during the review process.

This is important study, but has several concerns as mentioned by the reviewer. I hope you can revise the manuscript based on the comments.

We look forward to receiving your revised manuscript.

Kind regards,

Yoshio Yamaoka

Academic Editor

PLOS ONE

Journal Requirements:

2. We note that your paper includes detailed descriptions of individual patients/participants. As per the PLOS ONE policy (http://journals.plos.org/plosone/s/submission-guidelines#loc-human-subjects-research) on papers that include identifying, or potentially identifying, information, the individual(s) or parent(s)/guardian(s) must be informed of the terms of the PLOS open-access (CC-BY) license and provide specific permission for publication of these details under the terms of this license. Please download the Consent Form for Publication in a PLOS Journal (http://journals.plos.org/plosone/s/file?id=8ce6/plos-consent-form-english.pdf). The signed consent form should not be submitted with the manuscript, but should be securely filed in the individual's case notes. Please amend the methods section and ethics statement of the manuscript to explicitly state that the patient/participant has provided consent for publication: “The individual in this manuscript has given written informed consent (as outlined in PLOS consent form) to publish these case details

Reviewers' comments:

Reviewer's Responses to Questions

**Comments to the Author**

1. Is the manuscript technically sound, and do the data support the conclusions?

Reviewer #1: Yes

2. Has the statistical analysis been performed appropriately and rigorously? 

Reviewer #1: Yes

3. Have the authors made all data underlying the findings in their manuscript fully available?

Reviewer #1: Yes

4. Is the manuscript presented in an intelligible fashion and written in standard English?

Reviewer #1: Yes

5. Review Comments to the Author

Reviewer #1: In this study, Costa et al. have evaluated the expression of cathepsin S in gastric cancer and H. pylori infection. The study has reported excellent findings; however, before publication, several points deserve the authors’ attention.

Major comments

Abstract

1. Results. In the gastritis group, CTTS expression was significantly higher in patients with positive test for H. pylori than negative test for H. pylori: 87.5% and 38.5%, respectively (p<0.001).

Introduction

1. Paragraph 1. Move…… (cathepsin B, H, K, L, and S) after…….at least five (………) have ……..

2. Paragraph 2. Epidermal growth factor (EGF), vascular endothelial growth factor (VEGF), and tumor growth factor-beta (TGFβ),………….

3. Paragraph 2. Activated cathepsins hydrolyze growth factors, EGF, VEGF, and TGFβ, promoting cancer cell proliferation and angiogenesis,……….. The sentence is a bit confusing, after hydrolysis if they are destroyed then how cell proliferation occurs. The hydrolysis renders maturation of the EGF, VEGF, and TGFβ, then promotes the cancer cell proliferation.

Materials and Methods

1. Study design. Paragraph 1, line 7. ….. and other with a negative result for H. pylori (n=25).

2. For the sake of understanding it is better to name the tables in immunohistochemistry as Table 1, 2, and 3 and their title or it can be also expressed in text forms. The second table is the grading of CTTS stained cells of cancer and gastritis both or only tumor cells, not clear. In the demographic data table there are CTTS stained cells in both cancer and gastritis groups but in methods, you mentioned the grading of tumor cells.

Results

1. Table 1 title. Comparison between gastric cancer and gastritis groups (demographic data)

2. Table 1, Percentage of CTTS stained cells. In the methods, the percentage has been evaluated for cancer cells only but here in gastritis also. It needs clarification

3. Paragraph 3. The explanation in the paragraph does not match Table 3. Table 3 does not contain characteristics of gastric adenocarcinoma.

4. Paragraph 4. The explanation in paragraph and figure 1 legend does not corroborate. In the paragraph, it is mentioned gastritis whereas figure 1 legend depicts gastric adenocarcinoma.

5. Paragraph 5. Similar to the above comment, the explanation in paragraph 5 and figure 2 legend does not corroborate. In the paragraph, it is explained about gastric cancer and figure 2 legend it is about gastritis.

Discussions

1. Cathepsins that have previously………….. are B, E, K, L, S, X, and Z. Provide references

2. Paragraph 2, Last sentence. The results of these studies suggest………… Which studies you are referring is not clear.

3. Provide full form when you are referring first time and then short form; for example, provide full form of TNM

Minor comments

1. Three terms; gastric tumour, gastric cancer, and gastric adenocarcinoma have been used. I suggest using either one and making it uniform.

2. Throughout the manuscript, H. pylori should be italic

6. PLOS authors have the option to publish the peer review history of their article (what does this mean?). If published, this will include your full peer review and any attached files.

Reviewer #1: No

---

## [Author Response · Author response to Decision Letter 0]

23 Apr 2022

Dear Editors and Reviewers,

We would like to thank the reviewers for the constructive criticism regarding our manuscript entitled “Analysis of cathepsin S expression in gastric adenocarcinoma and in Helicobacter pylori infection”.

A point-by-point response to the reviewers’ comments was included below. The alterations performed in the manuscript are highlighted in yellow. All authors have read and approved the revised version of this manuscript. The authors have no financial interest and no conflicts of interest to disclose.

We look forward to hearing from you concerning on the suitability of the revised manuscript for publication.

Yours sincerely,

The corresponding author

# Reviewer 1

In this study, Costa et al. have evaluated the expression of cathepsin S in gastric cancer and H. pylori infection. The study has reported excellent findings; however, before publication, several points deserve the authors’ attention.

Answer: We thank the reviewer for the comments and constructive criticism. We tried to comply with all the suggestions made

Major comments

Abstract

1. Results. In the gastritis group, CTTS expression was significantly higher in patients with positive test for H. pylori than negative test for H. pylori: 87.5% and 38.5%, respectively (p<0.001).

Answer: We have adjusted the referred sentence.

Introduction

1. Paragraph 1. Move…… (cathepsin B, H, K, L, and S) after…….at least five (………) have ……..

Answer: We have adjusted the sentence.

2. Paragraph 2. Epidermal growth factor (EGF), vascular endothelial growth factor (VEGF), and tumor growth factor-beta (TGFβ),………….

Answer: We have adjusted the sentence according to what was indicated.

3. Paragraph 2. Activated cathepsins hydrolyze growth factors, EGF, VEGF, and TGFβ, promoting cancer cell proliferation and angiogenesis,……….. The sentence is a bit confusing, after hydrolysis if they are destroyed then how cell proliferation occurs. The hydrolysis renders maturation of the EGF, VEGF, and TGFβ, then promotes the cancer cell proliferation.

Answer: Cathepsins hydrolyze these growth factors, activating them and promoting cell proliferation. (Chen S, Dong H, Yang S, Guo H. Cathepsins in digestive cancers. Oncotarget. 2017;8(25):41690-41700. doi:10.18632/oncotarget.16677).

We have rephrased the sentence to turn it clearer.

Materials and Methods

1. Study design. Paragraph 1, line 7. ….. and other with a negative result for H. pylori (n=25).

Answer: We have adjusted the sentence.

2. For the sake of understanding it is better to name the tables in immunohistochemistry as Table 1, 2, and 3 and their title or it can be also expressed in text forms. The second table is the grading of CTTS stained cells of cancer and gastritis both or only tumor cells, not clear. In the demographic data table there are CTTS stained cells in both cancer and gastritis groups but in methods, you mentioned the grading of tumor cells.

Answer: We have removed the tables of the Methods section and put the information included in text format. 

The second table of IHC included both groups, gastritis and adenocarcinoma.

In the demographic.

In the demographic table, we included grading and expression of both groups. We have added this information in the Methods section.

Results

1. Table 1 title. Comparison between gastric cancer and gastritis groups (demographic data)

Answer: We have adjusted Table 1 title.

2. Table 1, Percentage of CTTS stained cells. In the methods, the percentage has been evaluated for cancer cells only but here in gastritis also. It needs clarification

Answer: We have adjusted this information in our Methods section.

3. Paragraph 3. The explanation in the paragraph does not match Table 3. Table 3 does not contain characteristics of gastric adenocarcinoma.

Answer: We have adjusted the paragraph according to Table 3. Thank you for pointing.

4. Paragraph 4. The explanation in paragraph and figure 1 legend does not corroborate. In the paragraph, it is mentioned gastritis whereas figure 1 legend depicts gastric adenocarcinoma.

Answer: We have adjusted the mentioned paragraph and figure. Thank you for pointing. We have changed figures order.

5. Paragraph 5. Similar to the above comment, the explanation in paragraph 5 and figure 2 legend does not corroborate. In the paragraph, it is explained about gastric cancer and figure 2 legend it is about gastritis.

Answer: We have adjusted the mentioned paragraph and figure. Thank you for pointing. We have changed figures order.

Discussions

1. Cathepsins that have previously………….. are B, E, K, L, S, X, and Z. Provide references

Answer: Reference #8 (Chen S, Dong H, Yang S, Guo H. Cathepsins in digestive cancers. Oncotarget. 2017;8(25):41690-41700. doi:10.18632/oncotarget.16677).

2. Paragraph 2, Last sentence. The results of these studies suggest………… Which studies you are referring is not clear.

Answer: We are referring to the results of our study and the previously reported reference earlier in the same paragraph, which are Liu et al. [14].

3. Provide full form when you are referring first time and then short form; for example, provide full form of TNM

Answer: The TNM staging system is a standardized nomenclature for the local and systemic staging for malignant tumours. The full form is the so called TNM.

Minor comments

1. Three terms; gastric tumour, gastric cancer, and gastric adenocarcinoma have been used. I suggest using either one and making it uniform.

Answer: We opted to use only gastric adenocarcinoma now.

2. Throughout the manuscript, H. pylori should be italic

Answer: We have adjusted it.

My co-authors and I found the recommendations of the reviewers to be extremely valuable and by addressing their concerns the manuscript has been significantly improved and we now hope that the manuscript is acceptable for publication. Thank you.

---

## [Editor Report · Decision Letter 1]

10 May 2022

Analysis of cathepsin S expression in gastric adenocarcinoma and in Helicobacter pylori infection

PONE-D-21-36174R1

Dear Dr. Adriano Carneiro Costa,

We’re pleased to inform you that your manuscript has been judged scientifically suitable for publication and will be formally accepted for publication once it meets all outstanding technical requirements.

Kind regards,

Yoshio Yamaoka

Academic Editor

PLOS ONE
---

## [Editor Report · Acceptance letter]

17 May 2022

PONE-D-21-36174R1 

Analysis of cathepsin S expression in gastric adenocarcinoma and in *Helicobacter pylori* infection 

Dear Dr. Costa:

I'm pleased to inform you that your manuscript has been deemed suitable for publication in PLOS ONE. Congratulations! Your manuscript is now with our production department. 

Kind regards, 

on behalf of

Professor Yoshio Yamaoka 

Academic Editor

PLOS ONE